

# Comparing transcriptome profiles of human embryo cultured in closed and standard incubators

Jingyu Li[1], Jiayu Huang[2], Wei Han[1], Xiaoli Shen[1], Ying Gao[2] and Guoning Huang[1]

[1] Chongqing Key Laboratory of Human Embryo Engineering, Chongqing Reproductive and Genetics Institute, Chongqing Health Center for Women and Children, Chongqing, China
[2] Department of Gynaecology and Obstetrics, Union Hospital, Tongji Medical College, Huazhong University of Science and Technology, Wuhan, China

## ABSTRACT

It is necessary to compare the transcriptomic profiles of human embryos cultured in time-lapse imaging (TLI) incubators and standard incubators (SI) in order to determine whether a closed culture system has a positive impact on embryos. In this study, we used RNA-sequencing (RNA-Seq) to characterize and compare the gene expression profiles of eight-cell embryos of the same quality grade cultured in TLI and SI. We sequenced a total of 580,952,620 reads for zygotes, TLI-cultured, and SI-cultured eight-cell embryos. The global transcriptomic profiles of the TLI embryos were similar to those of the SI embryos and were highly distinct from the zygotes. We also detected 539 genes showing differential expression between the TLI and SI groups with a false discovery rate (FDR) < 0.05. Using gene ontology enrichment analysis, we found that the highly expressed SI genes tended to execute functions such as transcription, RNA splicing, and DNA repair, and that the highly expressed TLI genes were enriched in the cell differentiation and methyltransferase activity pathways. This study, the first to use transcriptome analysis to compare SI and TLI, will serve as a basis for assessing the safety of TLI application in assisted reproductive technology.

## INTRODUCTION

Since the inception of clinical in vitro fertilization (IVF), assisted reproductive technology (ART) laboratories have focused on improving embryo culture systems. There have been significant developments in ART procedures over the past two decades, including the use of new culture medium (*Gardner & Lane, 1997*; *Mantikou et al., 2013b*; *Summers & Biggers, 2003*), the change from a 2 to 3 day culture duration to a 5–6 day culture duration (*Marek et al., 1999*; *Papanikolaou et al., 2006*), and the application of hypoxic medium that reportedly increases embryonic development and clinical outcomes (*Bontekoe et al., 2012*; *Kirkegaard, Hindkjaer & Ingerslev, 2013*; *Mantikou et al., 2013a*). Additionally, there has been a switch from static to dynamic culture systems, including tilting embryo culture, microfluidic culture, and mechanical vibration during culture (*Kim et al., 2009*;

Corresponding author
Guoning Huang,
gnhuang217@sina.com

*Matsuura et al., 2010*). Although numerous new ART techniques continue to be developed, proper validation of their safety is necessary.

Time-lapse imaging (TLI) is one of the most recently developed ART embryo culture systems. The TLI system permits the continuous evaluation of embryonic development and provides more information of the process (*Armstrong et al., 2015*; *Racowsky, Kovacs & Martins, 2015*). Several studies have claimed that the parameters obtained via TLI can be used for predicting the blastocyst formation, implantation, and pregnancy (*Cruz et al., 2012*; *Kaser & Racowsky, 2014*; *Kirkegaard et al., 2013*; *Meseguer et al., 2011*, *2012*; *Wong et al., 2010*; *Wu et al., 2016*). Furthermore, since TLI can assess embryos without removing them, embryos are not exposed to changes in temperature, humidity, light, gas concentrations, and pH, which can impair embryonic development (*Fujiwara et al., 2007*; *Zhang et al., 2010*). However, few studies have compared the performances of TLI culture systems to standard incubators (SI) (*Alhelou, Mat Adenan & Ali, 2018*; *Barberet et al., 2018*; *Sciorio, Thong & Pickering, 2018*; *Wu et al., 2017*). Our previous study showed that TLI had a higher transferable embryo ratio than the SI group on Day 3 (*Wu et al., 2017*), which was consistent with other studies (*Alhelou, Mat Adenan & Ali, 2018*; *Barberet et al., 2018*; *Sciorio, Thong & Pickering, 2018*). This suggests that there are differences between SI- and TLI-cultured embryos, indicating the importance of further exploring the molecular mechanisms and pathways underlying these variations.

Pre-implantation embryos are special cells that carry out the most dramatic genome-wide changes of gene expression (*Xue et al., 2013*; *Yan et al., 2013*). They are sensitive to their environment, particularly four- and eight-cell embryos in the process of embryonic genome activation (EGA) (*Deshmukh et al., 2011*; *Driver et al., 2012*; *Gad et al., 2012*; *Urrego et al., 2017*), and require accurate molecular regulation. Environmental changes disturb the normal gene expression that is essential for successful pre-implantation embryonic development (PED) (*Sirard, 2017*; *Urrego, Rodriguez-Osorio & Niemann, 2014*). For instance, oxygen concentrations outside the incubator can cause the decrease of *GLUT1*, *GLUT3*, and *VEGF* in embryos (*Kind et al., 2005*), suggesting that these genes are sensitive to environmental changes. Several studies have focused on improving culture conditions and reducing defects that might lead to transcription changes. RNA sequencing (RNA-Seq) can help determine whether there are differences in the transcriptomic profiles of human embryos cultured in TLI and SI.

In this study, we explored the influence of SI and TLI incubator culturing on the gene expression patterns of Day 3 embryos, and their subsequent influence on pathways and biological functions controlling human embryonic development. Our research provides the first comprehensive datasets for human eight-cell embryos cultured in SI and TLI using RNA-Seq, and the results will serve as a basis for assessing the safety of TLI application in ART.

# MATERIALS AND METHODS

## Ethics statement

This study was approved by the Institutional Review Board (IRB) of Chongqing Health Center for Women and Children (2016-RGI-01). We followed the guiding principles from

the Ministry of Science and Technology (MOST) for the review and approval of human genetic resources. All donor couples voluntarily donated embryos after signing written informed consent at the Chongqing Reproductive and Genetics Institute in the Chongqing Health Center for Women and Children.

## Intracytoplasmic sperm injection

We performed Intracytoplasmic sperm injection (ICSI) within 5 h of oocyte retrieval under an inverted microscope (Olympus IX70; Olympus Optical Co. Ltd., Tokyo, Japan) with a micromanipulation system (CellTram® 4r; Eppendorff, Hamburg, Germany). During ICSI, we placed the oocytes in pre-equilibrated culture droplets and covered them with six ml of mineral oil (Ovoil, Vitrolife, Gothenburg, Sweden). We used a holding pipette to position the oocyte. The sperm was injected into the cytoplasm with a micropipette when the first polar body reached the 6 or 12 o'clock position.

## Embryo culture

For the SI group, we placed the injected oocytes into a pre-equilibrated culture dish (Thermo Scientific, Waltham, MA, USA) with 25 μL of culture droplets (Vitrolife Sweden AB, Gothenburg, Sweden) covered with 1.2 mL of paraffin oil (Vitrolife Sweden AB, Gothenburg, Sweden). The embryos were cultured in an SI (MCO-5M; Sanyo, Osaka, Japan) at 37 °C with 5% $O_2$ and 6% $CO_2$ until embryo transfer on Day 3.

For the TLI group, we transferred the injected oocytes into pre-equilibrated Embryoslides and cultured them in the EmbryoScope$^{TM}$ until the time of transfer. The embryos cultured in the EmbryoScope$^{TM}$ had the same culture conditions as the SI group.

## Embryo scoring

We performed morphological assessments of the SI- and TLI-cultured embryos at the same time points using the same criteria. Additional information obtained during TLI was not used for embryo assessment or selection. In this study, the time points for morphological assessments followed the guidelines suggested by the ESHRE/Alpha consensus (31). The embryos were assessed at 17 ± 1, 44 ± 1, and 68 ± 1 h after ICSI. We scored embryos on their blastomere shape, blastomere number, and fragmentation rate. An embryo was defined as grade 1 when it had an even blastomere shape and <10% fragmentation, grade two when it had uneven blastomeres and 10–25% fragmentation, grade three when it had uneven blastomeres and 25–35% fragmentation, and not recommended for transfer when the fragmentation >35%. Generally, if one patient have more than eight embryos with grade 1–3, we will transfer two embryos with top quality, and freeze the other top six embryos. Then, the other embryos excluding the eight embryos were defined as not reach the transfer degree in this cycle, even if embryos with grade three. However, if one patient did not have enough embryos with grade one or two, for example, only one embryo with grade three. We will only transfer one embryo with grade three. Here, the embryos used in this study were all from the first situation, and were grade three with eight blastomere.

## Collecting human zygotes and eight-cell embryos

The patients who donated embryos were between 25 and 30 years old without a history of genetic diseases or smoking. We collected zygotes 6 h after ICSI, and eight-cell embryos on Day 3. Each couple donated only one zygote or embryo. To avoid possible embryo heterogeneity, each RNA-Seq pool had five zygotes or embryos from five different patients. We processed a total of 15 zygotes and 30 embryos (15 TLI-cultured and 15 SI-cultured) for RNA-Seq. All embryos were frozen using vitrification and were then stored.

The embryos were cultured for 2 h after thawing, briefly exposed to an acidic PBS solution for 5–10 s to remove the zona pellucida, and were washed three times in PBS. After all the embryos in one group were prepared, we removed them from the PBS, and immediately placed five embryos per tube into lysis buffer. All embryos used in this study exhibited normal morphology without reaching the transfer degree in this cycle.

## RNA sequencing library generation

We performed amplification using the Smart-seq2 method. We used the Qubit® 3.0 Fluorometer and Agilent 2100 Bioanalyzer to check the quality of the cDNA product and to ensure its length was around 1–2 kb. The library was prepared following the manufacturer's instructions (Illumina. Cat. FC-131-1024). After library preparation, we checked the library quality using the PerkinElmer LabChip® GX Touch and Step OnePlus™ Real-Time PCR System. The libraries were then sequenced on the Illumina HiSeq 4000 with a 150-bp paired-end.

## RNA sequence data processing

We used Trim_Galore to remove raw sequence reads that contained adapter contamination and poor-quality reads with low PHRED scores. Tophat (version 2.4.0) was used to map all data to the hg38 genome. We used RSEM (version 1.3.0) to calculate the gene expression levels based on the GRCh38.92 annotation database. We performed differential expression analysis using DESeq2, and only kept genes with significantly higher changes (log2 fold changes ≥1 and false discovery rates (FDR) <0.05). We deposited all RNA-Seq data in the NCBI Gene Expression Omnibus (GEO) under accession number GSE114771.

## Gene ontology analysis

We performed gene ontology (GO) analysis using the DAVID web-tool. Biological process, molecular function, and cellular component terms were selected as the background gene sets.

## RT-qPCR

Total RNA extraction, reverse transcription, and pre-amplification were performed using the Single Cell-to-CT qRT-PCR Kit (4458237; Invitrogen, Carlsbad, CA, USA) according to the manufacturer's instructions. We conducted real time PCR with SYBR Premix Ex Taq™ (TaKaRa, Kyoto, Japan) and the CFX96 Touch Real-Time PCR System (Bio-Rad, Hercules, CA, USA). We measured Ct values using Sequence Detection System software

**Table 1 Summary of the sequencing qualities and reads mapping.**

| Sample_ID | Raw reads | Clean reads (after QC) | Q30 (%) | Mapped reads | Rate (%) |
|---|---|---|---|---|---|
| Zygote #1 | 83,071,408 | 68,950,172 | 88.1 | 61,702,497 | 89.49 |
| Zygote #2 | 53,566,316 | 32,041,678 | 72.35 | 29,486,629 | 92.03 |
| Zygote #3 | 63,087,152 | 27,625,662 | 62.63 | 25,162,686 | 91.08 |
| Eight-cell SI #1 | 78,213,756 | 65,996,908 | 88.94 | 60,610,716 | 91.84 |
| Eight-cell SI #2 | 67,695,944 | 41,141,300 | 70.82 | 37,656,820 | 91.53 |
| Eight-cell SI #3 | 62,160,362 | 52,110,290 | 83.02 | 47,953,929 | 92.03 |
| Eight-cell TLI #1 | 67,586,738 | 53,242,892 | 80.21 | 49,691,133 | 93.33 |
| Eight-cell TLI #2 | 56,754,526 | 45,191,346 | 81.37 | 42,197,537 | 93.38 |
| Eight-cell TLI #3 | 48,816,418 | 39,410,428 | 81.86 | 36,581,706 | 92.82 |

(Bio-Rad, Hercules, CA, USA), and analyzed the gene expression level using the $2^{-\Delta\Delta Ct}$ method. All experiments contained at least three biological replicates. We used of the average of GAPDH and HPRT for normalizing RT-qPCR data. The primers used in this study are shown in Table S1.

## Statistical analysis

We used SPSS statistical software 19.0 for the statistical analyses, and one-way ANOVA to assess the differences between the two groups. For all analyses, differences were considered significant when $p < 0.05$.

# RESULTS

## Deep sequencing of the human embryo transcriptome

Using RNA-Seq technology, we studied the transcriptomic profiles of the three groups of human embryos: zygotes, SI-cultured, and TLI-cultured eight-cell embryos. After removing reads with adaptors, reads containing poly N, and low-quality reads from the raw data, a total of 68,950,172 (88.1% of the raw data), 32,041,678 (72.35%), and 27,625,662 (62.63%) trimmed reads remained in the zygote group; 65,996,908 (88.94%), 41,141,300 (70.82%), and 52,110,290 (83.02%) remained in the SI group; and 53,242,892 (80.21%), 45,191,346 (81.37%), and 39,410,428 (81.86%) remained in the TLI group, respectively (Table 1).

We analyzed the trimmed reads via TopHat for alignment with the reference genome (hg38). We mapped approximately 400 million paired-end reads derived from the nine samples, with a mapping rate ranging from 89% to 93% (Table 1). The typical number of detectable genes in the individual samples ranged from approximately 18,000 to 27,000 genes.

## Global transcriptomic profile analysis

The number of trimmed reads mapped to a gene was normalized to transcripts per kilobase million (TPM) values to accurately measure the gene expression level.
To determine whether the TLI- and SI-cultured embryos had distinct transcriptomic
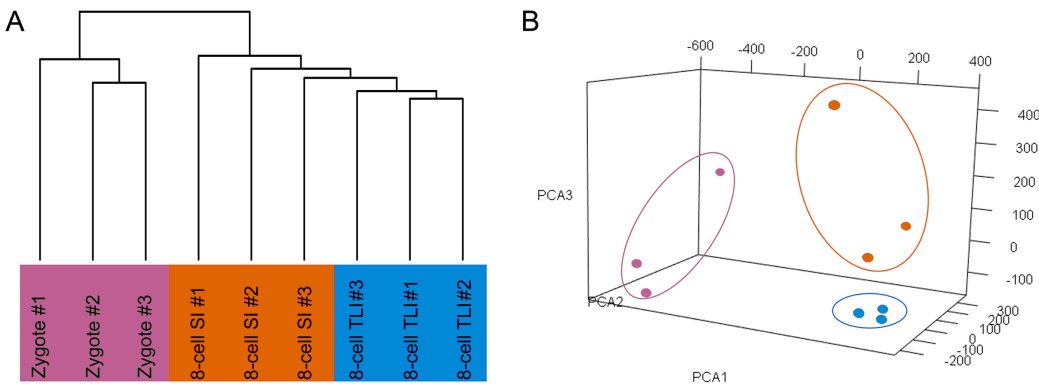

**Figure 1 Global analysis of expression patterns in human zygotes, and 8-cell embryos cultured in SI and TLI.** (A) Unsupervised hierarchical clustering. All genes expressed in at least one of the samples with TPM ≥ 0.1 were used for the analysis. (B) PCA of the transcriptome of human zygotes (pink), and 8-cell embryos cultured in TLI (blue) and SI (orange). PCA1, PCA2, and PCA3 represent the top three dimensions of the genes showing differential expression among these samples, which accounts for 32.7%, 14.5%, and 10.7% of the expressed genes, respectively.     

landscapes, we analyzed the RNA-Seq data using unsupervised hierarchical clustering. We included all genes that were expressed in at least one of the nine samples with TPM values ≥0.1. The results showed that embryos in the same developmental stage clustered together. Moreover, the TLI samples were similar to the SI samples and were distinct from the zygotes (Fig. 1A). Our principal component analysis (PCA) also found these distinct expression patterns (Fig. 1B).

## Differentially expressed gene analysis

Although the global transcriptomic profiles of the TLI group were more similar to those of the SI group than the zygotes, we could not distinguish any small molecular differences between TLI- and SI-cultured embryos. Therefore, we independently compared the TLI and SI groups and identified a total of 539 differentially expressed genes (DEGs, FDR <0.05, >2-fold change). We verified the accuracy using cluster analysis, which successfully distinguished the TLI from the SI group (Fig. 2A). Of the 539 DEGs, 24 genes were exclusively expressed in the TLI group, and 39 genes were exclusively expressed in the SI group. Of the remaining 476 genes, 188 showed higher expression in the TLI group while 288 genes showed higher expression in the SI group (Table S2). We also detected the differential expression of solute carrier family (SLC) genes, including *SLC5A12*, *SLC9B2*, and *SLC35A3*, which play important roles in early embryonic development (*Song et al., 2005*). Figure 2B shows the fold-change breakdown across the DEGs of the two groups.

## Functional analysis

To identify the functions of DEGs, we performed GO enrichment analysis of SI- and TLI-overexpressed genes. In SI-overexpressed genes, we found many more enriched GO terms, particularly biological process terms (83%, Fig. 3). The overexpressed genes in SI tended to execute functions, such as transcription, RNA splicing, and DNA repair. However, the overexpressed genes in TLI belonged to several different categories,

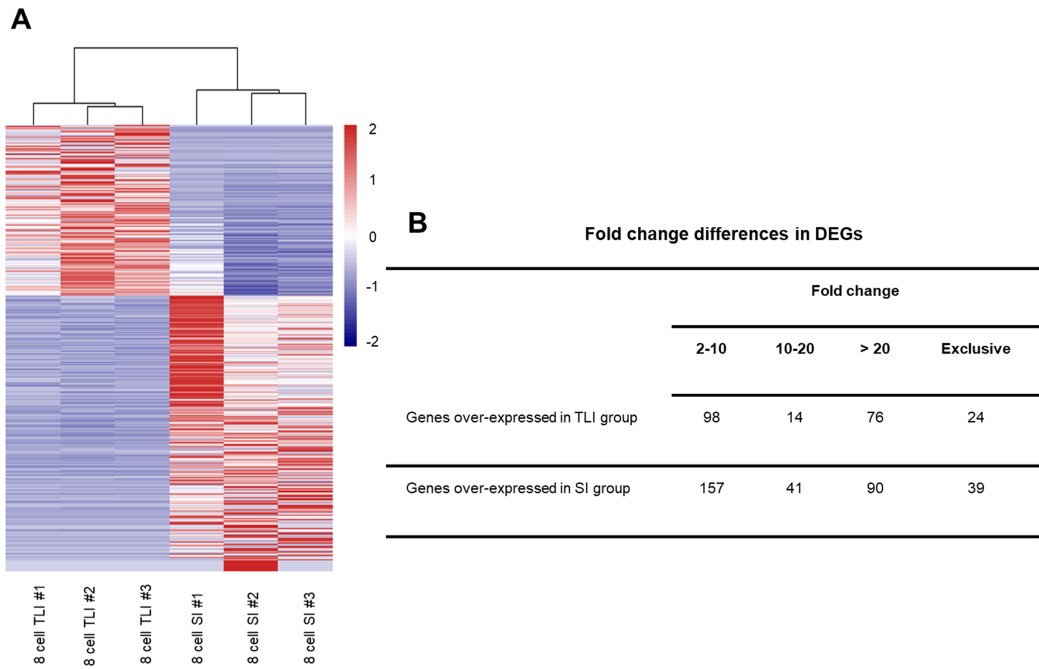

**Figure 2 The differentially expressed genes between SI- and in TLI-cultured embryos.** (A) Heatmap and hierarchical cluster of DEGs. (B) Fold-change differences in expression of DEGs.

including the cell differentiation and methyltransferase activity pathways. In spite of the similarities between the SI and TLI groups, there were still small differences that could influence specific pathways and molecular functions during embryonic development.

## Validating differentially expressed genes

To verify the reliability of our sequencing data, we selected four upregulated and downregulated genes from several enriched pathways, including RNA splicing, DNA repair, cell differentiation, and methyltransferase activity. We used qPCR to analyze their expression levels in eight-cell embryos cultured in SI and TLI. We detected the upregulated expressions of *RBM14*, *TPRAP*, *SMARCB1*, and *ELAVL1* in SI, and *ANGPT1*, *SLC35A3*, *PRDM5*, and *ZIC3* in TLI (Fig. 4). These results were consistent with our sequencing data.

## DISCUSSION

There has been a longstanding debate regarding the superiority of the closed culture system over the SI system. To settle this debate, the safety of the closed TLI system first needs to be confirmed. Previous studies that have compared TLI and SI focused on embryonic development and clinical outcomes (*Barberet et al., 2018*; *Chen et al., 2017*; *Kaser & Racowsky, 2014*; *Meseguer et al., 2012*; *Park et al., 2015*; *Wu et al., 2017*; *Wu et al., 2016*). However, none have explored whether TLI and SI culturing results in transcriptomic alterations in embryos. We performed the first transcriptomic analysis comparing the gene expression profiles of SI- and TLI-cultured human embryos.

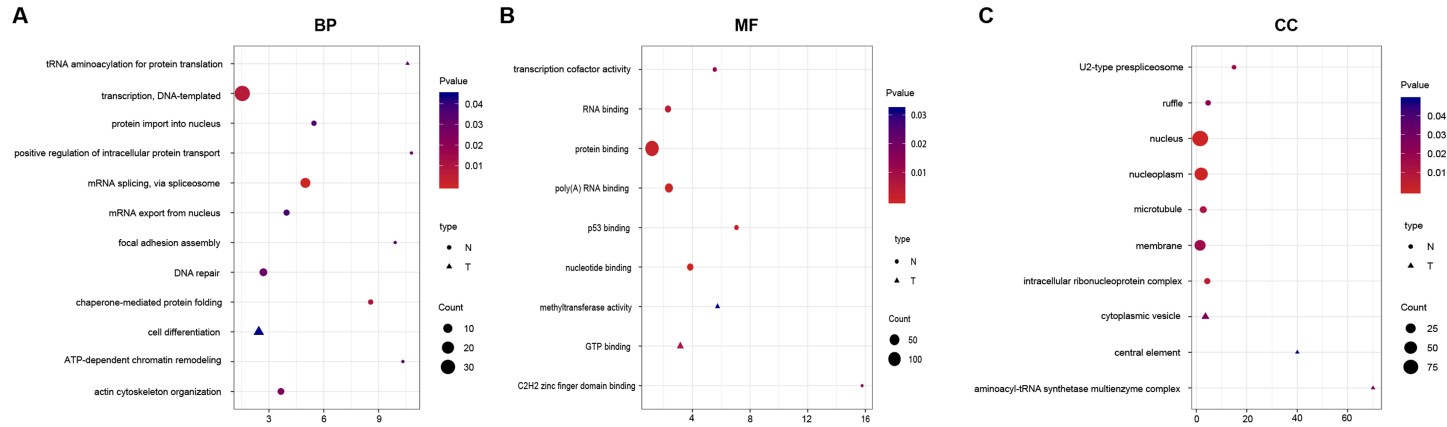

**Figure 3 Function enrichment analysis result of DEGs.** The colors indicate the significance (*p*-value), the size represents the number of genes enriching the corresponding annotation, and the shape shows the sample group. The fold enrichment of analysis is shown in the horizontal axis. (A) "Biological process" term. (B) "Molecular function" term. (C) "Cellular component" term.

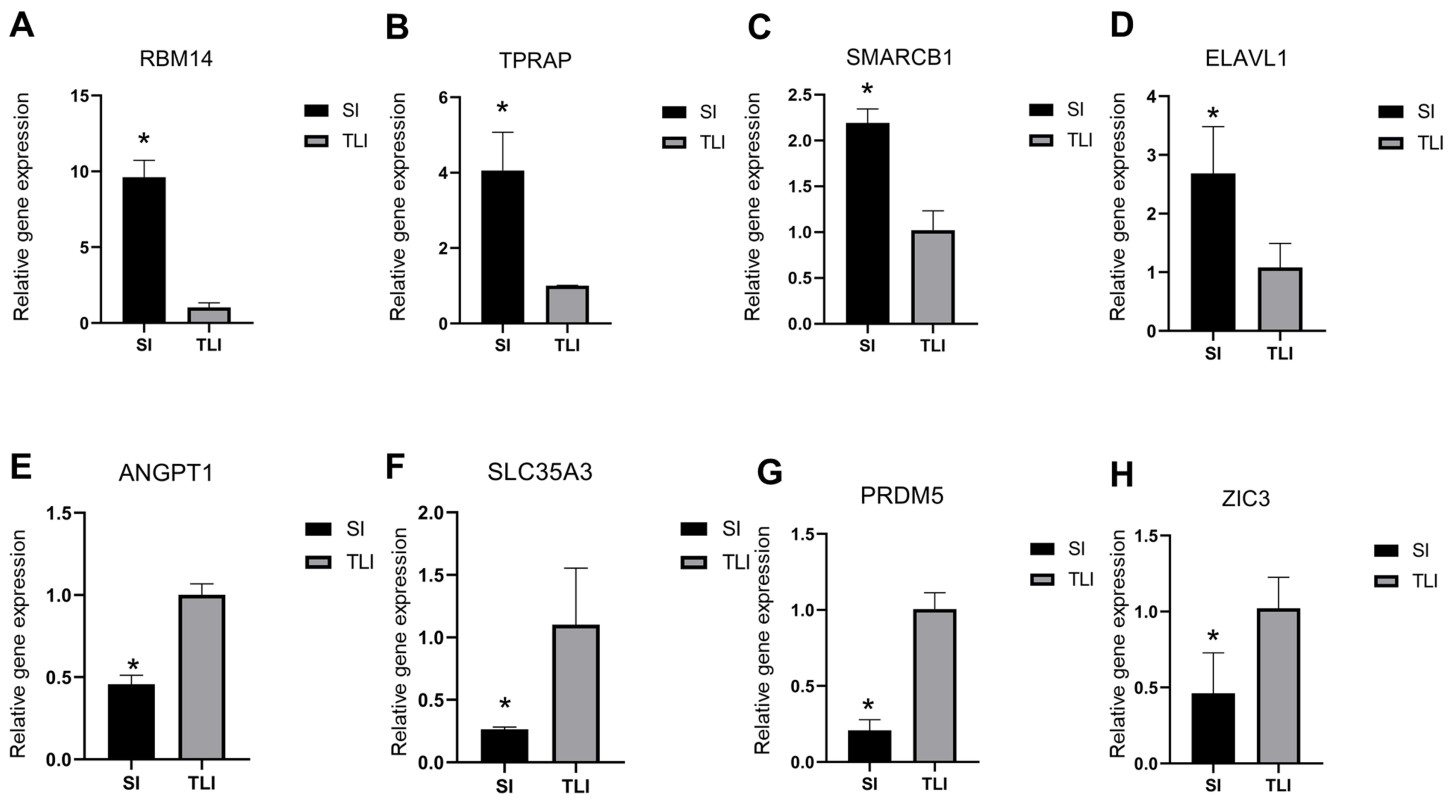

**Figure 4 qPCR validation of the eight differentially expressed genes.** (A–D) The upregulated expression level of embryos in SI were confirmed by qPCR. (E–H) The upregulated expression level of embryos in TLI were confirmed by qPCR. The average of the GAPDH and HPRT was used for normalization. Results are presented as mean values ± SEM. *Indicate significant differences (*p* < 0.05).

Our results showed similarities between the global transcriptomic profiles of the SI and TLI groups.

Time-lapse imaging has two main advantages: (i) it can provide a more stable in vitro culture environment for embryo development, and (ii) using continuous imaging, TLI

offers embryologists more embryo morphokinetic parameters which can improve the efficiency of embryo selection in ART. In this study, we only focused on the first advantage. To accurately compare the two culture systems, we selected embryos of equivalent morphological quality in the TLI and SI groups. The embryos were cultured in individual medium droplet. Additionally, we did not use the morphokinetic parameters provided by TLI for embryo assessment, and we assessed the embryos in both groups at the same time points using the same criteria.

Embryos are sensitive to microenvironmental changes during in vitro culture. Embryonic developmental kinetics, transcription, proteomes, metabolic state, histone remodeling, and methylation patterns have been shown to change when exposed to atmospheric oxygen (*Harvey et al., 2004*; *Katz-Jaffe et al., 2005*; *Li et al., 2016*; *Takahashi et al., 2000*; *Van Soom et al., 2002*). Opening the incubator door disrupts the $CO_2$ tension of the embryo culture microenvironment, temporarily changing the pH of the culture medium and negatively affecting the embryo. A previous study reported that during SI culturing, temperature recovery took approximately 30 min after a 5 s door opening/ closing procedure, and the recovery of oxygen tension took about 7.8 ± 0.9 min (*Fujiwara et al., 2007*). Exposure to visible light can also create additional stress for embryos (*Ottosen, Hindkjaer & Ingerslev, 2007*). Nonetheless, the morphological assessment of embryos is a vital step in a typical ART program. This method requires moving embryos outside the incubator for microscopic observation once daily from Day 1 to Day 3 after fertilization. Most laboratories generally culture 10 dishes in one SI, so there are at least 30 door opening/closing procedures when SI culturing. The dish is removed only once in TLI. *Zhang et al. (2010)* reported that decreasing the number of embryo observation times could significantly improve the blastocyst development rate. Although TLI allows for the continuous imaging of embryonic development in an undisturbed culture system, the long exposure to light during TLI might affect normal gene expression in embryos.

Although the two embryo groups were of similar morphological grades, they may have had different transcriptomes. Since our primary objective was to investigate the differences between the transcriptomic landscapes of TLI- and SI-cultured embryos, we pooled five embryos in one sample to balance the heterogeneity from different patients. In total, we identified 539 genes showing differential expression, and 284 genes with greater than 10-fold differences in gene expression level. We distinguished these two groups using hierarchical clustering, which showed that the embryos from the two groups had small molecular differences. We hypothesized that the transcriptional changes in the embryos were caused by a consistent directional response to the different culture microenvironments.

In addition to the individual gene differences, our study also revealed several significantly enriched GO terms. One of particular interest was the RNA splicing pathway, which contains 14 genes with high expression in the SI group. Alternative splicing is one type of gene regulatory mechanism at the RNA level that ensures proper transcriptome regulation in PED, especially during EGA. Our previous study showed that the overexpression of *HNRPU*, which acts as an alternative splicing regulator, can affect the embryonic development of mice (*Wang et al., 2016*). We found that SI-cultured embryos

displayed a higher expression of *HNRPUL1*, a *HNRPU*-like gene. The gene *ELAVL1*, which can bind to AU-rich genes including SLC genes and whose activity is regulated by the p38 MAPK pathway, also showed high expression in the SI group (*Calder, Watson & Watson, 2011*; *Fan & Steitz, 1998*; *Peng et al., 1998*; *Song et al., 2005*). Previous studies have found that the inhibition of the p38 MAPK pathway can block PED and decrease the expression of SLC genes (*Song et al., 2005*). In accordance with these results, we found downregulation of several SLC genes, including *SLC5A12*, *SLC9B2*, and *SLC35A3*, in the SI group. We postulated that the different environments in SI may have disrupted normal alternative splicing regulation during human EGA. In the SI group, we found higher expressions of other genes involved in DNA repair, including *DOT1L*, *MMS19*, *POLL*, *SMARCB1*, *POLE*, *CEP164*, *TRRAP*, and *RBM14*. The activation of these genes might have been linked to oxidative stress or an aberrant DNA damage response, both of which affect long-term embryo viability (*Ooga, Suzuki & Aoki, 2013*). There is evidence that a low expression of *DOT1L* is required for PED and heterochromatin remodeling (*Phillips, Wildt & Comizzoli, 2016*; *Tao et al., 2017*). Additionally, we detected a few genes in the TLI group with high expressions of *ACTN1*, *MAP1A*, *CFL1*, *FLNB*, *DNASE1*, *WHAMM*, *PFN1*, and *SYNE2* related to the light stimulus response despite the lack of significant enrichment. Although it has been reported that exposure to light during TLI observations does not affect PED (*Nakahara et al., 2010*), the potential long-term effects have not been studied. Our results showed that in spite of their similarities, the differences between the two groups could affect particular pathways and important molecular functions of the embryos.

The limitations of this study are worth mentioning. The embryos used in this study were not top grade, and it is known that early embryos can self-correct during subsequent development (*Coticchio et al., 2019*). Therefore, future studies should further explore the differences between blastocysts cultured in SI and TLI.

## CONCLUSION

This is the first study to compare the transcriptome of embryos cultured in closed TLI and SI. Although the global transcriptomic profiles were similar between the SI and TLI groups, their distinct culture microenvironments produced several different results. The impact of these small differences in gene expression on embryonic development requires further research. Our study provides a basis to understanding the molecular mechanisms underlying these differences, and represents a new form of research on the safety of TLI in ART.

### Funding

This study was supported by the Special fund for clinical research of Chinese Medical Association (17020430712, 18010260755), and Chongqing YuZhong Science Project (20170127). The funders had no role in study design, data collection and analysis, decision to publish, or preparation of the manuscript.

## Grant Disclosures

The following grant information was disclosed by the authors:
Chinese Medical Association: 17020430712 and 18010260755.
Chongqing YuZhong Science Project: 20170127.

## Competing Interests

The authors declare that they have no competing interests.

## Author Contributions

- Jingyu Li conceived and designed the experiments, performed the experiments, analyzed the data, prepared figures and/or tables, authored or reviewed drafts of the paper, and approved the final draft.
- Jiayu Huang performed the experiments, authored or reviewed drafts of the paper, and approved the final draft.
- Wei Han performed the experiments, authored or reviewed drafts of the paper, and approved the final draft.
- Xiaoli Shen analyzed the data, authored or reviewed drafts of the paper, and approved the final draft.
- Ying Gao conceived and designed the experiments, authored or reviewed drafts of the paper, and approved the final draft.
- Guoning Huang conceived and designed the experiments, authored or reviewed drafts of the paper, and approved the final draft.

## Human Ethics

The following information was supplied relating to ethical approvals (i.e., approving body and any reference numbers):

The Institutional Review Board (IRB) of Chongqing Health Center for Women and Children approved this research (2016-RGI-01).

## Data Availability

Data is available at NCBI GEO: GSE114771.

## Supplemental Information

Supplemental information for this article can be found online at http://dx.doi.org/10.7717/peerj.9738#supplemental-information.

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
