# Peer review of "Comparing transcriptome profiles of human embryo cultured in closed and standard incubators"

_PeerJ, doi:10.7717/peerj.9738_

## Round 0.1 · original submission · Major Revisions

Both of the reviewers and I feel that there is great potential in this paper. However, you will note that both reviewers felt that the data must be validated by RT-qPCR for publication in PeerJ (or any other substantive journal). While we appreciate that these samples are precious, the data cannot fully be interpreted without such validation.

We hope that you will consider this revision and resubmit your manuscript after completion of these studies.

Reviewer 1 ·

Basic reporting

1) The grammar, spelling mistakes and references in the manuscript need to be checked:

a. Line 30: We is spelled wo
b. Line 50: change provide to provides
c. Line 55: reference missing
d. Line 59: Replace changing to changes
e. Line 91: Change u to µ
f. Line 120: Change zygote to zygotes
g. Line 131: Please check cDNA production. Do you mean cDNA product?
h. Line 172: Give full form for TPM abbreviation
i. Line 193: Change group to groups, add a coma
j. Line 222: change PH to pH
k. Line 227: change require to requires
l. Line 237-239: Please check grammar
m. Line 244: Check grammar
n. Line 246: says genes play important roles but no mention of what roles and how it affects the embryos
o. Line 259: Change significantly to significant
p. Line 260: Change have to has
q. Line 270: Inconsistent fonts
r. Line 274: Change difference to differences, have to has and add an
s. Line 275: Change need to needs
t. Line 449: Check grammar
u. Line 453: Change represent to represents

Experimental design

The main focus of this manuscript was to compare the transcriptomic profiles of 8-cell embryos cultured in standard incubator (SI) to time-lapse imaging incubator (TLI) using RNA-sequencing. Authors found that the global transcriptomic profiles of the 8-cell embryos were similar except for 539 genes that were differentially expressed. Using GO enrichment analysis authors found the different functional pathways that those genes were part of.
Overall, this is an interesting study because it is imperative to confirm that the newer ART/IVF laboratory materials and methods not only have no detrimental effects but also improve the quality of the embryos. RNA-sequencing is a great tool to test the differences between gene expression profiles of these embryos cultured in two different incubators.

1) Section: Methods, Subsection: Embryo score. The embryos were scored on a scale of 1-3. Which ones were chosen for RNA-sequencing?

2) Section: Methods, Subsection: Embryo score. For the SI, how often were the embryos taken out for observation? Did that match the usual methodology for embryos cultured in SI?

Validity of the findings

1) Although RNA-sequencing generates a lot of useful data, it is incomplete without proper validation. The authors have mentioned this, however, this approach makes the manuscript purely descriptive.

2) The functions/pathways of the over expressed genes in both groups are generated through bioinformatics analysis. The manuscript should address why they might see the differences in the 2 groups with regards to the cellular pathways and what that might mean for the quality of the embryos. This should be addressed in the discussion.

3) The paper could be elevated if there is mention of any data that may have looked at the differences between the embryo qualities or even blastocyst rates between the 2 incubators. This can confirm whether functionally, one method might be better than the other. This may also address the differences in gene expression profiles.

Reviewer 2 ·

Basic reporting

There are multiple grammatical issues throughout the manuscript that need addressing, particularly with respect to tense, when to use singular versus plural, and overall sentence structure. A native English speaker should review the manuscript beforehand.

Experimental design

Please see below regarding transfer degree, embryo pooling, morphokinetic measurements, and reporting of raw data.

Validity of the findings

The authors state that the validation of the RNA-seq data by quantitative RT-PCR (RT-qPCR) was not performed due to human embryos being precious and that there is no solid evidence that this “provides extra significance to the results.” It’s fairly well-accepted that RT-qPCR should be conducted regardless due to errors that RNA-seq platforms can produce and that the sensitivity of the RNA-seq data is based on the read depth. Therefore, RT-qPCR of leftover RNA or RNA isolated from new embryos should be completed to confirm the findings.

Additional comments

The main objective of this study by Li and colleagues was to examine the transcriptomic profiles of human embryos cultured in a time-lapse imaging (TLI) incubator and a standard incubator (SI) to determine whether a closed culture system has a positive impact on embryo development. Using RNA-sequencing (RNA-seq) of pools of zygotes and TLI versus SI 8-cell embryos of similar morphological grade, the authors determined that the whole transcriptomes of TLI and SI embryos were similar, but distinct from that of zygotes. More importantly, they show that there was differential expression of 539 genes in TLI and SI 8-cell embryos. Gene Ontology (GO) analysis identified higher expression of genes involved in cell differentiation and DNA methylation in TLI embryos, whereas genes associated with RNA transcription, splicing, and DNA repair were enriched in SI embryos. Overall, the manuscript is not well written and there are multiple issues that need to be addressed in order to support the authors’ conclusions of the study as follows:

Major:

(1) The authors state that “All human embryos used in this study exhibited normal morphology, however didn’t reach the transfer degree.” This is contradictory and additional morphological details with regards to what constitutes the transfer degree and whether there were differences in the number of embryos with different transfer degrees between embryo groups needs to be clarified. Under the limitations in the Discussion section, it also needs to be stated that all the human embryos used in this study were not of the highest quality and therefore, their findings require further investigation.

(2) It is stated that 5 embryos per group were pooled for RNA-seq analysis. Why was this done when there are several low-input sequencing protocols available, including those for even single cells? How can the authors discount possible embryo heterogeneity within the same or different pools? Also, how many patients are represented in each embryo group to ensure the results are not patient specific?

(3) Besides providing a closed culture system, TLI incubators enable the measurement of morphokinetic parameters and yet, the authors did not do this. Since the embryos were pooled for RNA-seq analysis, were there any differences in mitotic timing or cytokinesis (bipolar versus multipolar divisions) between embryos within the same pool or different pools? The findings of the study would be more impactful if these analyses were included.

(4) The authors report the percentage of raw sequencing data that remained after trimming and quality control assessment. It’s the average genome coverage that should be calculated instead and based on this coverage, will determine whether the authors actually performed “deep sequencing.”

(5) The name of the differentially expressed genes are identified for the first time in the Discussion. This data to should be moved to the Results section in a logical manner to support the GO findings.

(6) From the methods section, it sounds as if the zona pellucida was removed and the embryos transferred to lysis buffer after freeze-thaw and pooling. Is this accurate? There are no methodological details provided for ICSI.

Minor:

(1) The use of the term “sequencing fragments” could be confused with cellular fragments. I would suggest that the authors be more specific in their language.
(2) On lines 90 and 95, the authors refer to “injected oocytes,” but it’s not until line 119 that the term ICSI is first introduced.
(3) On line 117, the authors state that the embryos were collected from “volunteers.” I think the more appropriate term is “patients.” Also, does this age range refer to maternal age only? Typically, the average age is given. The authors do not state the total number of patients that donated embryos.

---

## Round 0.2 · Minor Revisions

As you will note, the reviewers find the modified manuscript much improved and have suggested only minor changes. Please address the textual changes and perform a careful reading of the text for grammatical and linguistic errors. Also, pay careful attention to address Reviewer 2's comments about experimental design and validity and address these issues in the text.

Reviewer 1 ·

Basic reporting

The re-submitted version of the article submitted by Li et al has improved dramatically . The major issues with the previous version have been addressed by the authors.
Major Comment: It would be helpful to have some more specific functions of upregulated genes in the discussion section.
Minor Comments:
Line 65: delete 'of'
Line 169: check grammar
Line 176: check grammar
Line 180: check grammar
Line 311: change exposure to exposed
Line 322: change cultured to culture
Line 355: Change leaded to lead
Line 362: Missing citation/ reference
Line 384: Check grammar
Line 385: Check grammar
Figure 4: Label Y axis

Experimental design

In the previous version 2 of the major concerns were 1) English, and 2) lack of qPCR validation of the upregulated genes. Both have been addressed.

Validity of the findings

This study is impactful since it gives a snapshot of differentially expressed genes in the embryos cultured in the SI and TLI system.

Additional comments

All the previous concerns have been addressed in the rebuttal.

Reviewer 2 ·

Basic reporting

(1) There are still numerous grammatical issues throughout the manuscript that need addressing. For example, see:
(a) Line 125: “more information of on the process”
(b) Line 183: “which in turn lead”
(c) Line 276: “ICSI was performed with 5 h of oocytes retrieval”
(d) Lines 331-332: “without family heredity case history and smoke history.”
(e) Lines 333-453: “Every RNA-Seq pools have 5 zygotes or embryos from 5 different patients, which can discount possible embryo heterogeneity from patient factor.”
(f) Line 454: “were vitrification frozen”
(g) Line 458: “immediately every five embryos each tubes.”
(h) Lines 459-460: “present study exhibited normal morphology, however didn’t reach the transfer degree in this cycle.”
(i) Line 652: “human embryos transcriptomes”
(j) Line 828: “especially the biological processes terms”
(k) Line 918: “changed when exposure”
(l) Line 1021: “possessed several different transcriptome.”
(m) Line 1170: “which contain”
(n) Line 1176: “leaded to the blockage”
(o) Lines 1183-1184: “were several evidence showed”
(p) Line 1226: “needs to further research”

Experimental design

(2) The authors do not specify if a DNase step was included or exon-spanning primers used to remove contaminating genomic DNA (gDNA) and avoid amplification of gDNA, respectively, for quantitative RT-PCR (RT-qPCR).

(3) A minimum of 2 genes, preferably more, should be used for the normalization of RT-qPCR data in order to avoid individual gene variation between embryo groups. This is particularly important since the authors use GAPDH to normalize all the RT-qPCR data, but GAPDH has been shown to be more highly expressed in aneuploid human embryos (Vera-Rodriguez et al. Nat Comm 2015) and aneuploidy is common in cleavage-stage human embryos.

(4) Additional experimental details are needed in the Materials and Methods section even if described previously. For ICSI, the manipulator system used. For RNA-seq, the Illumina kit used (75, 150, 300 bp?). I assume paired-end?

Validity of the findings

(5) The word “clean” is not typically used to describe sequencing data after processing. A more appropriate term is “trimmed” or “processed.”

(6) On Lines 1218-1220, the authors state: “In addition, it is well known that early embryos have the ability of correct the alterations in gene expression during the subsequent development.” Do you have a reference for this statement b/c as far as I know this has not been established.

Additional comments

In the revised version of their manuscript, Li and colleagues addressed some of my questions and concerns. However, there are still certain issues that need to be remedied.

---

## Round 0.3 · Minor Revisions

I have made comments directly on the rebuttal letter in RED about the lack of addressing reviewer 2's editorial and scientific issues raised in the last review. While there may have been attempts to address the editorial issues, many still remain in the sections Reviewer 2 highlighted. The rest of the manuscript should be read again by the English language editor since clearly the myriad of issues that remain.

Both the editorial and scientific issues need to be addressed for the manusript to be acceptable for publication.

---

## Round 0.4 · Minor Revisions

The manuscript is much improved but needs two small edits:

1) Please add the reference to the comment at the end that "embryos can self correct"

2) The details of the Embryo Scoring (note, not "Embryo Score") provided in the rebuttal are more detailed than added to the Methods. Please elaborate more specifically in the Methods (but check the grammar as there are many errors in the rebuttal letter section).

---

## Round 0.5 · accepted · Accept

Thank you for the hard work in improving the manuscript through several revisions. I hope that you found the process improved the science and its ability to be communicated to other researchers. We hope you will consider PeerJ for other manuscripts in the future.